# Visceral Pleural Invasion as a Determinant of Surgical Strategy in Non–Small Cell Lung Cancer: A Multicenter Study

**DOI:** 10.3390/cancers17203382

**Published:** 2025-10-20

**Authors:** Wakako Nagase, Yujin Kudo, Takuya Nagashima, Takahiro Mimae, Yoshihisa Shimada, Masaru Hagiwara, Masatoshi Kakihana, Tatsuo Ohira, Yoshihiro Miyata, Hiroyuki Ito, Morihito Okada, Norihiko Ikeda

**Affiliations:** 1Department of Surgery, Tokyo Medical University, 6-7-1 Nishishinjuku, Shinjuku-ku, Tokyo 160-0023, Japan; nagase@tokyo-med.ac.jp (W.N.); zenkyu@tokyo-med.ac.jp (Y.S.); masaru-h@tokyo-med.ac.jp (M.H.); m-kaki@tokyo-med.ac.jp (M.K.); tohira@tokyo-med.ac.jp (T.O.);; 2Department of Thoracic Surgery, Kanagawa Cancer Center, 2-3-2 Nakao, Asahi, Yokohama 241-8515, Japan; naga16@kcch.jp (T.N.); h-ito@kcch.jp (H.I.); 3Department of Surgical Oncology, Hiroshima University, 1-2-3 Kasumi, Minami-ku, Hiroshima 734-8551, Japan; tmimae@hiroshima-u.ac.jp (T.M.); ymiyata@hiroshima-u.ac.jp (Y.M.); morihito@hiroshima-u.ac.jp (M.O.)

**Keywords:** VPI, pathological pleural invasion, lymph node metastasis, non-small-cell lung cancer

## Abstract

**Simple Summary:**

In the current era, sublobar resection is increasingly utilized for the treatment of small-sized lung cancers, raising the question of how to best evaluate risk factors that influence long-term outcomes. Visceral pleural invasion (VPI) has been regarded as an adverse prognostic factor in non-small-cell lung cancer (NSCLC) for decades, but its clinical significance is now being fundamentally reassessed. A key question remains whether VPI simply reflects tumor size or instead represents a distinct pattern of metastatic behavior. In this retrospective multicenter study of over two thousand patients, we found that VPI was strongly linked to higher rates of lymph node involvement and distinct patterns of spread, leading to worse survival outcomes. By contrast, tumors with lepidic growth rarely demonstrated VPI or nodal spread. These results suggest that VPI appears to be aggressive tumor biology and should be carefully considered when determining the extent of lung and lymph node resection, especially as surgical strategies evolve toward more limited procedures.

**Abstract:**

Background: Visceral pleural invasion (VPI) has traditionally been regarded as a negative prognostic indicator in non-small-cell lung cancer (NSCLC). However, with the increasing adoption of sublobar resection for small-sized NSCLC, the clinical significance of VPI is being fundamentally reassessed. Specifically, it remains uncertain whether VPI is indicative of tumor size or represents distinct metastatic behavior. Methods: We conducted a retrospective comprehensive multicenter study involving 2464 patients with pathologically confirmed NSCLC ≤ 3 cm who underwent complete resection at three Japanese institutions. The prevalence, metastatic patterns, and prognostic impact of VPI were systematically evaluated, with particular focus on histological growth patterns. Results: VPI was identified in 370 patients (15%). Notably, VPI-positive tumors demonstrated a doubled incidence of lymph node metastasis (31% vs. 15%, *p* < 0.001) and a distinct metastatic profile characterized by preferential hilar spread (#12, 16.9%) and an increased risk of skip N2 metastasis (4.0% vs. 2.0%). Five-year recurrence-free survival was significantly reduced in the VPI group (33.7% vs. 50.6%, respectively). Conversely, adenocarcinomas with lepidic characteristics demonstrated a minimal risk of VPI or nodal metastasis, with incidences of 2% and 1%, respectively. This finding highlights the heterogeneity in the biological aggressiveness of small-sized NSCLC. Conclusions: Our findings suggest that VPI is not merely a histopathological descriptor but also acts as a clinically significant indicator of aggressive metastatic behavior, potentially enhancing surgical and staging approaches beyond just considering tumor size. With the increasing adoption of sublobar resection for small-sized NSCLC, recognizing that VPI appears to be associated with predominant hilar involvement and an elevated risk of skip N2 metastasis may help refine decisions on the extent of lung and lymph node resection.

## 1. Introduction

Lung cancer remains one of the leading causes of cancer-related mortality worldwide, with prognosis remaining generally unfavorable [1]. Despite recent advances in perioperative systemic therapies that have raised expectations for improved outcomes, even small, early-stage non-small-cell lung cancer (NSCLC) can harbor occult lymph node metastasis [2,3]. Recent advancements in the surgical management of early-stage NSCLC have led to a paradigm shift from lobectomy to sublobar resection for small tumors. Trials such as JCOG0802 and CALGB140503 have demonstrated that segmentectomy offers outcomes comparable to lobectomy for tumors ≤2 cm, while JCOG1211 supported its safety and efficacy for peripheral adenocarcinomas ≤3 cm with ground-glass nodule-predominant features [4,5,6]. Consequently, the proportion of patients undergoing sublobar resection has rapidly increased in appropriately selected cases.

While these findings are encouraging, it is important to recognize that not all small tumors exhibit indolent characteristics [3,7]. Some small NSCLC tumors harbor aggressive pathological features and present with early recurrence or distant metastasis following sublobar resection. This underscores the clinical need to identify high-risk tumors accurately, even when radiological evaluations suggest early-stage disease. Among such risk factors, visceral pleural invasion (VPI) has emerged as a particularly important prognostic marker. VPI is associated with increased rates of lymph node metastasis and reduced survival, even in tumors measuring less than 3 cm [8,9].

VPI is more frequently observed in higher-grade adenocarcinomas, whereas it is rarely seen in well-differentiated tumors [10]. As tumor grading through imaging or biopsy becomes more widespread, the ability to predict VPI preoperatively is improving [11]. High-resolution computed tomography (CT) and artificial intelligence (AI)-based tools have enhanced the prediction of tumor invasiveness by incorporating features such as the consolidation-to-tumor ratio, solid component size, and radiomic texture [12,13]. These technologies may support the identification of tumors at high risk for VPI, even before or during surgery. Such information can inform surgical decision-making, including whether to pursue sublobar resection or lobectomy.

The present study aims to comprehensively evaluate the pathological relationship between VPI, histologic subtypes—including the predominant subtype of adenocarcinoma—and the pattern of lymph node metastasis in surgically resected NSCLC tumors measuring ≤3 cm. By elucidating the clinicopathological characteristics associated with VPI, these findings contribute to a deeper understanding of its prognostic impact on small tumors and provide valuable insights for improving risk prediction and guiding treatment decisions in early-stage lung cancer.

## 2. Materials and Methods

### 2.1. Patient Cohort

This retrospective, multicenter, observational study utilized data from the HITOKA3 project database. A total of 2464 patients with surgically resected NSCLC tumors measuring 3 cm or less in pathological whole tumor size were included between January 2010 and December 2019. Data were collected from three institutions, Hiroshima University, Kanagawa Cancer Center, and Tokyo Medical University. Patients were excluded if they had tumors larger than 3 cm, a pathological diagnosis of small-cell lung cancer, or had undergone procedures other than lobectomy or segmentectomy. Wedge resections were not included in this analysis. Mediastinal lymph node dissection (ND2a) was the standard procedure in both lobectomy and segmentectomy. Intraoperative frozen-section examination of suspicious hilar or mediastinal nodes was routinely performed, and when negative, mediastinal dissection could be curtailed at the surgeon’s discretion according to the surgical strategy of each institution. All patients underwent preoperative CT and ^18^F-fluorodeoxyglucose positron emission tomography/computed tomography (PET/CT). Clinical staging was performed according to the eighth edition of the TNM staging system [14], and patients were reclassified based on the ninth edition criteria [15]. Collected clinicopathological data included age, sex, smoking history, clinical stage, type of surgical procedure, type of lymph node dissection, number of resected lymph nodes, number of metastatic lymph nodes, histologic type, pathological stage, pleural invasion, and prognosis. The study protocol was approved by the institutional review board at Tokyo Medical University (SH2969), which waived the requirement for informed consent due to the retrospective nature of the study.

### 2.2. Patient Follow-Up

Based on the attending physician’s discretion, patients typically received routine follow-up examinations at intervals of approximately three to six months during the first two years and annually thereafter. Follow-up evaluations included physical examination, chest radiography, CT, and tumor marker measurements. Recurrence was diagnosed through physical examination and diagnostic imaging of suspected lesions. When recurrence was suspected, additional evaluations such as whole-body CT, brain magnetic resonance imaging, PET/CT, or bone scintigraphy were conducted.

### 2.3. Pathological Classification

Pathological staging was performed according to the eighth edition of the TNM classification, and histopathological analyses were conducted based on the fifth edition of the World Health Organization (WHO) classification system [14,16]. Cases were reclassified according to the ninth edition of the TNM staging system [15]. VPI was defined as tumor invasion beyond the elastic layer of the pleura, classified into four categories: PL0 (no invasion beyond the elastic layer), PL1 (invasion beyond the elastic layer but not reaching the pleural surface), PL2 (invasion to the pleural surface), and PL3 (invasion into the parietal pleura or chest wall). This standardized classification was applied to all surgically resected tumors in this study [16]. Hematoxylin and eosin staining and Elastica van Gieson staining were routinely used to evaluate histological architecture and pleural invasion. All resected lymph nodes were classified according to the nodal map established by the International Association for the Study of Lung Cancer [17]. Skip N2 metastasis was defined as pathological involvement of mediastinal lymph nodes (pN2a or pN2b) in the absence of hilar lymph node metastasis (pN1).

### 2.4. Statistical Analysis

Overall survival (OS) was defined as the interval between the date of surgery and the date of death or last follow-up. Recurrence-free survival (RFS) was defined as the interval between the date of surgery and the date of initial recurrence, death, or last follow-up. The Mann–Whitney U test was used for continuous variables, and Pearson’s chi-square test was applied to categorical variables to compare adenocarcinoma (LUAD) versus non-LUAD groups, as well as VPI-positive versus VPI-negative groups. A *p*-value of less than 0.05 was considered statistically significant. Kaplan–Meier survival curves were constructed to estimate survival rates. Survival differences between patient groups categorized by pathological lymph node metastasis and VPI status were compared using the log-rank test. Multivariate analyses of factors associated with lymph node metastasis were performed using logistic regression. Covariates included in the multivariable analysis were age, sex, smoking history, tumor location, pathological tumor size, histologic type, and VPI status.

Statistical analyses were performed using the Statistical Package for the Social Sciences (SPSS) software (version 29.0; SPSS Inc., Chicago, IL, USA) and R (version 4.2.2).

## 3. Result

### 3.1. Patient Characteristics

Patient characteristics, including a comparison by histological type, are summarized in Appendix A. The study included 2464 patients: 1298 men (52.7%) and 1166 women (47.3%), with a median age of 69 years (range, 20–93 years). Lobectomy was performed in most cases (73.1%), and segmentectomy in 26.9%. The extent of lymph node dissection varied by surgical procedure, with mediastinal lymphadenectomy more frequently performed in lobectomy cases (83.8%) than in segmentectomy cases (43.3%). Histological analysis revealed LUAD in 2114 patients (85.8%). The median pathological tumor size was 2 cm (range, 0.2–3 cm). VPI was observed in 370 patients (15.0%), and lymph node metastasis in 296 patients (12.0%). Tumor recurrence occurred in 274 patients (11.1%). In this cohort, VPI-positive tumors occurred more often in men and smokers, while age distributions were similar between groups.

### 3.2. Difference in Histological Subtype and VPI

Detailed histological subtypes and their relationship with VPI are presented in Appendix A. Among the 2114 patients with LUAD, 893 (42.2%) had subtypes classified as adenocarcinoma in situ (AIS), minimally invasive adenocarcinoma (MIA), or lepidic adenocarcinoma (Lep). VPI was significantly less frequent in these subtypes, with only 17 cases (1.9%) showing VPI. Lymph node metastasis was observed in just 6 patients (0.7%) in this group. These findings indicate a low prevalence of VPI in specific LUAD subtypes, suggesting a lower risk of aggressive tumor behavior.

### 3.3. Prognostic Factors Related to VPI and Survival Analysis

To evaluate the impact of VPI on prognosis, patients with LUAD subtypes known to have favorable outcomes—AIS, MIA, and Lep—were excluded (see the flow chart in Figure 1). The resulting analysis included 1571 patients, of whom 353 (22.4%) were VPI-positive. Prognostic factors associated with VPI in this cohort are summarized in Table 1. VPI was significantly associated with an increased prevalence of lymph node metastasis (VPI-positive vs. VPI-negative: 31.4% vs. 14.9%, *p* < 0.001). This association remained significant in patients with tumors smaller than 2 cm (VPI-positive vs. VPI-negative: 23.9% vs. 11.2%, *p* < 0.001) (see Table 2). In terms of predominant subtypes, although micropapillary adenocarcinoma is recognized as an aggressive subtype, the rates of VPI and lymph node metastasis in micropapillary adenocarcinoma were comparable to those in the overall cohort (VPI-positive: 40% vs. VPI-negative: 20%, *p* = 0.349). The 5-year OS was significantly lower in the VPI-positive group (78.3% vs. 88.1%, *p* < 0.001). Similarly, the 5-year RFS was worse in the VPI-positive group (61.0% vs. 88.0%, *p* < 0.001). Figure 2 and Figure 3 display survival curves based on VPI and lymph node metastasis. Among patients without lymph node involvement, the 5-year OS was 90.8% for those without VPI and 82.3% for those with VPI (log-rank *p* < 0.001). In patients with lymph node metastasis, there was no significant difference in survival between VPI groups; the 5-year OS was 70.8% for patients without VPI and 68.0% for those with VPI (log-rank *p* = 0.759). The median overall survival time was not reached in any group. Tumor recurrence was observed in 264 patients (16.8%). Among node-negative patients, the 5-year RFS rate was significantly lower in those with VPI (72.5%) compared to those without VPI (94.1%) (log-rank *p* < 0.001). In the node-positive group, the 5-year RFS rate was further reduced: 50.6% in patients without VPI and 33.7% in patients with VPI (log-rank *p* = 0.005). In patients with tumors <2 cm, there was no significant difference in overall survival between those with and without VPI (5-year OS: 85.1% vs. 89.1%, log-rank, *p* = 0.128). In contrast, among patients with tumors 2 to 3 cm, survival was significantly worse in those with VPI compared to those without VPI (5-year OS: 73.1% vs. 86.8%, log-rank, *p* < 0.001) (see Appendix A). Regarding recurrence-free survival, patients with tumors <2 cm with VPI had significantly worse outcomes compared to those without VPI (5-year RFS: 75.0% vs. 90.5%, log-rank, *p* < 0.001). Similarly, in tumors 2 to 3 cm, recurrence-free survival was significantly poorer in patients with VPI than in those without VPI (5-year RFS: 50.7% vs. 84.7%, log-rank, *p* < 0.001) (see Appendix A). Univariate and multivariate analyses of factors associated with lymph node metastasis are summarized in Table 3. In univariate analysis, sex, pathological tumor size, and VPI were significant prognostic factors. Multivariate analysis identified VPI as an independent prognostic factor, along with sex, tumor location, and pathological tumor size (adjusted odds ratio = 2.24, 95% CI: 1.68–2.98, *p* < 0.001). The narrow confidence interval indicates sufficient precision of estimation, which was considered an alternative to a priori sample size calculation in this retrospective study. To further validate the findings, we additionally performed multivariate analysis, including the extent of lymph node dissection, and confirmed that VPI remained an independent prognostic factor; the results were consistent.

### 3.4. VPI and Areas of Lymph Node Metastasis

A detailed analysis of the distribution of lymph node metastases and their association with VPI was conducted on 1562 patients, excluding 9 patients with missing lymph node data (Table 4). Both N1 and N2 lymph node metastases were significantly more common in patients with VPI (*p* < 0.001). In the VPI-positive group, N2 metastasis was more frequent than N1 metastasis (VPI-positive vs. VPI-negative: 16.0% vs. 14.9%). Multistation N2 (N2b) metastases were also more frequently observed in the VPI-positive group (5.1% vs. 1.2%). Skip N2 metastasis was higher in the VPI-positive group as well (4.0% vs. 2.0%, *p* = 0.003). For N1 lymph node metastases, the lobar zone (#12) showed the highest rate, with an overall frequency of 10.1%, increasing to 16.9% in VPI-positive patients. Appendix A provide further details on the distribution of lymph node metastases in relation to histological subtype and tumor location.

## 4. Discussion

This study comprehensively evaluated the clinicopathological significance of VPI in small-sized NSCLC, focusing on its association with histologic subtypes, tumor grade, and lymph node metastasis. Sublobar resection has become increasingly common for early-stage NSCLC, particularly for tumors measuring ≤3 cm in recent years [4,5,6]. VPI is a known poor prognostic factor in lung cancer and plays a key role in predicting postoperative recurrence. Our findings confirm that VPI is significantly associated with worse RFS and OS, even in tumors ≤3 cm. VPI-positive tumors demonstrated significantly higher rates of lymph node metastasis, involving both N1 and N2 stations, along with a notably increased incidence of skip N2 metastasis. In our cohort, NSCLC tumors ≤3 cm—particularly LUAD with lepidic growth—exhibited a lower incidence of VPI. When excluding low-grade histologic subtypes, patients with VPI-positive tumors had a 31% incidence of lymph node metastasis and a recurrence rate exceeding one-third. These findings suggest that the presence of VPI may warrant more extensive surgical management, such as systematic lymphadenectomy or conversion to lobectomy, even in small, peripheral tumors.

Previous pathological studies have shown that VPI is more frequently observed in tumors with high-grade patterns, such as solid and micropapillary components, and is rarely seen in well-differentiated, lepidic-predominant tumors [18,19,20]. Nitadori et al. investigated the correlation between early-stage LUAD and VPI in node-negative LUAD measuring less than 3 cm [19]. Their findings indicated that VPI had limited significance in the lepidic subtype of LUAD and emphasized that the frequency of VPI varies by histological subtype. Our results were consistent with those of previous studies, as we also observed a low VPI frequency in in situ, minimally invasive, and lepidic subtypes of LUAD. Recent advances have enabled the preoperative estimation of tumor grade using high-resolution computed tomography and AI through metrics such as the consolidation-to-tumor ratio and solid component size [21]. The concordance of histological grades between preoperative biopsy and surgical specimens exceeds 80%, particularly for grade 1 and grade 3 adenocarcinomas [22]. Intraoperative frozen section analysis has demonstrated high accuracy in predicting invasiveness in tumors ≥1 cm, thereby aiding in the prevention of inadequate resections [23], underscoring its clinical utility. These diagnostic tools may improve surgical planning by identifying tumors at risk for VPI prior to or during surgery. Recent investigations have focused on the preoperative and intraoperative prediction of VPI using imaging modalities and AI-based approaches [24,25,26,27,28,29,30]. Takizawa et al. reported thoracoscopic diagnostic accuracies for VPI of 56.5% using conventional white light and 76.0% with autofluorescence imaging [24]. Shimada et al. further demonstrated improved diagnostic performance (73.9–78.2%) through deep learning analysis of thoracoscopic tumor images, surpassing the diagnostic accuracy of experienced thoracic surgeons (60.8–71.7%) [25]. In our institutional study, an AI model trained on clinical and imaging data achieved a predictive accuracy of 76.4% for VPI, aligning with previous findings [26]. These findings highlight the importance of accurate pathological evaluation and the continued development of imaging technologies to improve diagnostic precision. Such advances suggest that reliable risk stratification for VPI may soon be achievable even before or during surgery.

Previous studies have examined the adverse impact of VPI in small-sized tumors. A recent report by Altorki et al. presented a secondary analysis of the randomized clinical trial CALGB140503, focusing on patients with clinically node-negative NSCLC tumors smaller than 2 cm that were pathologically upstaged to T2 based on the presence of VPI [31]. The 5-year RFS was 58.2% in the VPI-positive group compared to 73.1% in the VPI-negative group, indicating unexpectedly high recurrence rates—including distant recurrences—and poorer survival outcomes in small, peripheral NSCLCs with VPI. Interestingly, in that cohort, there were no significant survival differences between surgical procedures (lobectomy vs. segmentectomy). In our study, patients with VPI had a higher recurrence rate (34.6%) than those without VPI (11.7%). These findings are consistent with the results of CALGB140503, as the 5-year RFS rate was significantly lower in the VPI-positive group (61.0%) compared to the VPI-negative group (88.0%).

Previous studies have reported an association between VPI and lymph node metastasis. Adachi et al. evaluated the impact of VPI on postoperative locoregional recurrence in patients with N1 disease [24]. In a subgroup analysis of 69 patients with N1 status, the presence of VPI was associated with poor survival outcomes [32]. Regarding the incidence of lymph node metastasis in VPI-positive tumors, several reports have shown that tumors ≤3 cm with VPI—particularly those ≤2 cm—are associated with more frequent lymph node involvement and poorer prognosis [9]. The association between VPI and lymph node metastasis is believed to be related to the dense lymphatic network of the visceral pleura, which facilitates the spread of tumor cells through the subpleural lymphatics to the hilar and mediastinal lymph nodes, and ultimately into the cervical venous circulation [8,9]. These studies, along with our findings, underscore the correlation between VPI and lymph node metastasis, as well as the poor prognosis associated with small-sized lung cancers exhibiting VPI. Although similar investigations have been conducted, our study further examined the prognostic impact of VPI by excluding cases with histological subtypes characterized by a low prevalence of VPI, to clarify its independent prognostic significance.

In addition, we analyzed the anatomical distribution of lymph node metastases and found a significantly higher incidence of skip N2 metastasis in VPI-positive patients. Similarly, Gorai et al. identified VPI as a significant factor associated with skip N2 metastasis [33]. As discussed previously, this is believed to result from the lymphatic channels that provide a direct route from the visceral pleura to the mediastinal lymph nodes. Our study also demonstrated a higher incidence of multistation N2 (N2b) metastases in VPI-positive patients compared to VPI-negative patients. Given the recent updates in the TNM classification, which emphasize the distinction between single and multiple N2 metastases, these findings provide valuable insight into the relationship between VPI and the revised staging system.

There are several limitations to this study. First, it was retrospective in design, and variations in surgical procedures across participating institutions may have influenced the findings. Differences in lymph node dissections between lobectomy and segmentectomy could have affected the evaluation of lymph node metastasis, potentially leading to underestimation in certain cases. Second, in assessing pathological factors related to VPI, we included various histological subtypes of lung cancer without accounting for tumor location. Since VPI is more frequently observed in peripheral tumors, its incidence and associated lymph node metastasis rates may differ in a cohort limited to peripheral tumors. Third, our study focused primarily on the relationship between VPI and pathological characteristics. Although recent advances in imaging technology have demonstrated the potential for preoperative VPI prediction, radiological tools for this purpose are not yet standardized in clinical assessment. Despite these limitations, one of the key strengths of this study is the use of the large and well-characterized HITOKA3 project database, which includes 2464 patients and permits detailed subgroup analysis of biologically aggressive tumors. The integration of the most recent TNM and WHO classifications, along with comprehensive anatomical lymph node mapping, strengthens the clinical relevance and applicability of the findings. We believe that these results will help identify high-risk cases and contribute to improved outcomes for patients with small tumors. Future prospective studies are warranted to confirm the associations identified between VPI, lymph node metastasis, and skip N2 metastasis, and to elucidate their clinical implications for surgical decision-making.

## 5. Conclusions

Our findings indicate that LUAD with lepidic growth has a lower incidence of VPI. Despite the small tumor size, VPI-positive cases exhibited a 31% incidence of lymph node metastasis, with metastases frequently involving the hilar region. Recent advances in lung cancer imaging, including AI-assisted analysis, have enhanced the ability to predict VPI preoperatively. These findings provide important insights into the association between VPI and lymph node metastasis, thereby supporting improved perioperative risk stratification and informing surgical strategies in early-stage lung cancer.

## Figures and Tables

**Figure 1 cancers-17-03382-f001:**
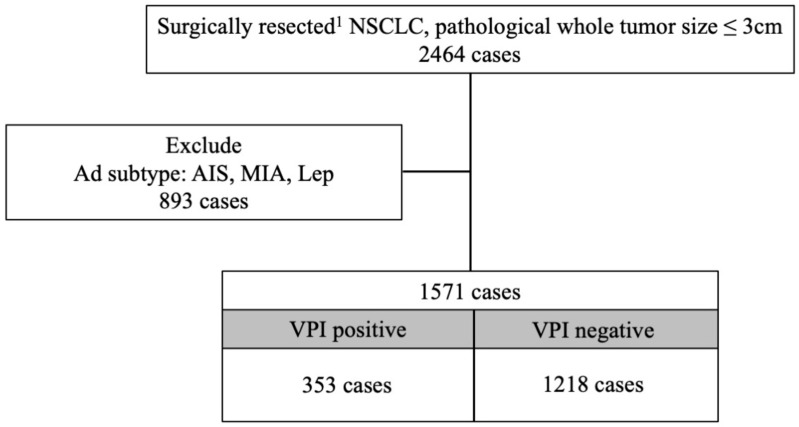
Flow chart of patients. A total of 1571 cases were analyzed, excluding 893 cases based on histological subtype: adenocarcinoma in situ (AIS), minimally invasive adenocarcinoma (MIA), or lepidic adenocarcinoma (Lep). ^1^ Lobectomy or segmentectomy; NSCLC, Non-small-cell lung cancer; AIS, adenocarcinoma in situ; MIA, minimally invasive adenocarcinoma; Lep, lepidic adenocarcinoma.

**Figure 2 cancers-17-03382-f002:**
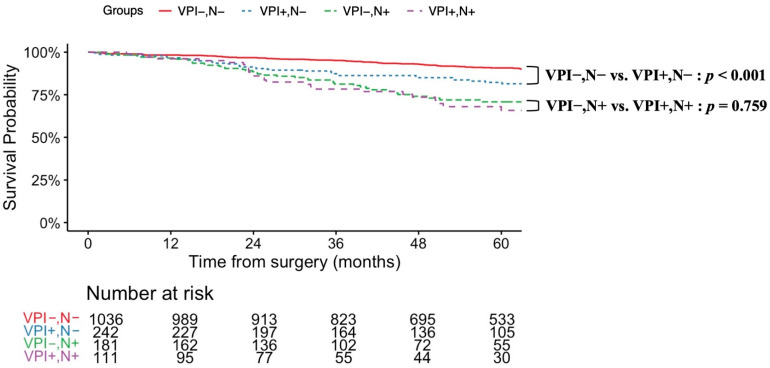
Overall survival curves based on VPI and lymph node metastasis. Overall survival curve excluding patients with histological subtypes: adenocarcinoma in situ (AIS), minimally invasive adenocarcinoma (MIA), or lepidic adenocarcinoma (Lep). In node-negative patients (*n* = 1278), the 5-year OS was significantly higher in those without VPI (*n* = 1036, 90.8%) compared to those with VPI (*n* = 242, 82.3%) (log-rank, *p* < 0.001). Among node-positive patients (*n* = 292), there was no significant difference in OS between VPI groups: 181 cases (70.8%) without VPI vs. 111 cases (68.0%) with VPI (log-rank *p* = 0.759). VPI, visceral pleural invasion; N, lymph node.

**Figure 3 cancers-17-03382-f003:**
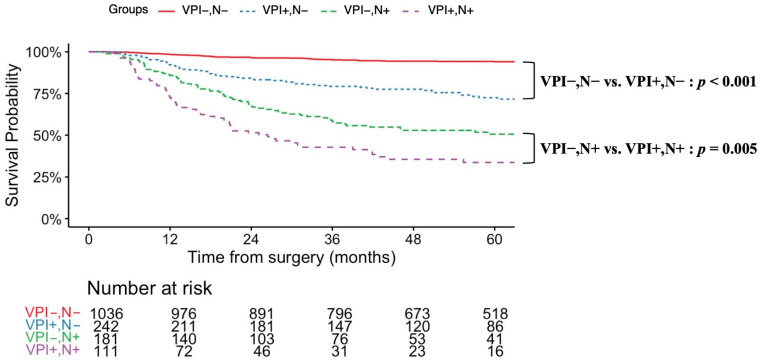
Recurrence-free survival curves based on VPI and lymph node metastasis. Recurrence-free survival curve excluding patients with histological subtypes: adenocarcinoma in situ (AIS), minimally invasive adenocarcinoma (MIA), or lepidic adenocarcinoma (Lep). Among node-negative patients (*n* = 1278), the 5-year RFS was 94.1% in those without VPI (*n* = 1036) and 72.5% in those with VPI (*n* = 242). In node-positive patients (*n* = 292), the 5-year RFS decreased to 50.6% without VPI (*n* = 181) and 33.7% with VPI (*n* = 111). VPI, visceral pleural invasion; N, lymph node.

**Table 1 cancers-17-03382-t001:** Pathologic prognostic factors of 3 cm tumors, related to VPI. Excluded cases of Adenocarcinoma subtypes of in situ, minimally, lepidic.

Variable	Overall, *n* = 1571	Non-VPI, *n* = 1218	VPI, *n* = 353	*p*-Value (Non-VPI vs. VPI)
**Age, Median (Minimum, Maximum)**	69 (62, 75)	69 (63, 75)	68 (61, 75)	0.067
**Sex, *n* (%)**				<0.001 ^1^
Men	945 (60.2)	702 (57.6)	243 (68.8)	
Women	626 (39.8)	516 (42.4)	110 (31.2)	
**Smoker, *n* (%)**	1025 (65.3)	766 (63.0)	259 (73.4)	<0.001 ^1^
**CT component of tumor, *n* (%)**				<0.001 ^1^
Solid	1018 (64.8)	739 (60.7)	279 (79.0)	
Part solid	496 (31.6)	429 (35.2)	67 (19.0)	
Pure GGO	27 (1.7)	27 (2.2)	0 (0.0)	
**Histological type**				0.233 ^1^
Adenocarcinoma	1221 (77.7)	952 (78.2)	269 (76.2)	
Squamous cell carcinoma	238 (15.1)	187 (15.4)	51 (14.4)	
Others	112 (7.1)	79 (6.5)	33 (9.3)	
**Surgical procedure, *n* (%)**				<0.001 ^1^
Segmentectomy	320 (20.4)	277 (22.7)	43 (12.2)	
Lobectomy	1251 (79.6)	941 (77.3)	310 (87.8)	
**Tumor location, *n* (%)**				0.546 ^1^
Left	627 (39.9)	491 (40.3)	136 (38.5)	
Right	944 (60.1)	727 (59.7)	217 (61.5)	
**Lymph node dissection, *n* (%)**				<0.001 ^1^
Not conducted	35 (2.2)	31 (2.5)	4 (1.1)	
ND1	240 (15.3)	196 (16.1)	44 (12.5)	
ND2a-1	1077 (68.6)	844 (69.3)	233 (66.0)	
ND2a-2	219 (13.9)	147 (12.1)	72 (20.4)	
**Median pathological tumor size, (Minimum, Maximum)**	2.0 (0.2, 3.0)	2.00 (0.2, 3.0)	2.2 (0.8, 3.0)	<0.001 ^1^
**Pathological T factor, 9th edition, *n* (%)**				
T1	1196 (76.1)	1196 (98.2)	0 (0.0)	
T2	312 (19.9)	12 (1.0)	300 (85.0)	
T3	58 (3.7)	10 (0.8)	48 (13.6)	
T4	5 (0.3)	0 (0.0)	5 (1.4)	
**Pathological N factor, 9th edition, *n* (%)**				<0.001 ^1^
0	1278 (81.4)	1036 (85.1)	242 (68.6)	
Lymph node metastasis	292 (18.6)	181 (14.9)	111 (31.4)	
N1	156 (9.9)	102 (8.4)	54 (15.3)	
N2a	102 (6.5)	64 (5.3)	38 (10.8)	
N2b	32 (2.0)	14 (1.1)	18 (5.1)	
NA ^2^	3 (0.2)	2 (0.2)	1 (0.3)	
**Pathological stage, 9th edition, *n* (%)**				<0.001 ^1^
0	3 (0.2)	3 (0.2)	0 (0.0)	
ⅠA1	173 (11.0)	173 (14.2)	0 (0.0)	
ⅠA2	595 (37.9)	595 (48.9)	0 (0.0)	
ⅠA3	252 (16.0)	252 (20.7)	0 (0.0)	
ⅠB	214 (13.6)	6 (0.5)	208 (58.9)	
IIA	97 (6.2)	97 (8.0)	0 (0.0)	
ⅡB	147 (9.4)	70 (5.7)	77 (21.8)	
ⅢA	67 (4.3)	20 (1.6)	47 (13.3)	
ⅢB	19 (1.2)	0 (0.0)	19 (5.4)	
ⅣA	1 (0.1)	0 (0.0)	1 (0.3)	
Unknown	3 (0.2)	2 (0.2)	1 (0.3)	
**Recurrence, *n* (%)**	264 (16.8)	142 (11.7)	122 (34.6)	<0.001

^1^ Wilcoxon rank sum test; Pearson’s Chi-squared test; Fisher’s exact test. ^2^ NA includes two cases with unclassifiable N2 disease and one case with NX. VPI, Visceral pleural invasion; GGO, Ground Glass Opacity; NA, Not applicable.

**Table 2 cancers-17-03382-t002:** Pathologic prognostic factors of 2 cm tumors, related to VPI. Excluded cases of Adenocarcinoma subtypes of in situ, minimally, lepidic.

Variable	Overall, *n* = 842	Non-VPI, *n* = 687	VPI, *n* = 155	*p*-Value (Non-VPI vs. VPI)
**Surgical procedure, *n* (%)**				<0.001 ^1^
Segmentectomy	239 (28.4)	212 (30.9)	27 (17.4)	
Lobectomy	603 (71.6)	475 (69.1)	128 (82.6)	
**Tumor location, *n* (%)**				0.617 ^1^
Left	330 (39.2)	272 (39.6)	58 (37.4)	
Right	512 (60.8)	415 (60.4)	97 (62.6)	
**Lymph node dissection, *n* (%)**				0.057 ^1^
Not conducted	24 (2.9)	22 (3.2)	2 (1.3)	
ND1	159 (18.9)	135 (19.7)	24 (15.5)	
ND2a-1	569 (67.6)	465 (67.7)	104 (67.1)	
ND2a-2	90 (10.7)	65 (9.5)	25 (16.1)	
**Median pathological tumor size, (Minimum, Maximum)**	1.5 (0.2, 2.0)	1.5 (0.2, 2.0)	1.6 (0.8, 2.0)	0.084 ^1^
**Pathological T factor, 9th edition, *n* (%)**				<0.001 ^1^
T1	684 (81.2)	684 (99.6)	0 (0.0)	
T2	130 (15.4)	1 (0.1)	129 (83.2)	
T3	26 (3.1)	2 (0.3)	24 (15.5)	
T4	2 (0.2)	0 (0.0)	2 (1.3)	
**Pathological N factor, 9th edition, *n* (%)**				<0.001 ^1^
0	727 (86.3)	609 (88.6)	118 (76.1)	
Lymph node metastasis	112 (13.3)	76 (11.1)	36 (23.2)	
N1	65 (7.7)	45 (6.6)	20 (12.9)	
N2a	39 (4.6)	28 (4.1)	11 (7.1)	
N2b	8 (1.0)	3 (0.4)	5 (3.2)	
NA	3 (0.4)	2 (0.3)	1 (0.6)	
**Pathological stage, 9th edition, *n* (%)**				<0.001 ^1^
0	3 (0.4)	3 (0.4)	0 (0.0)	
IA1	160 (19.0)	160 (23.3)	0 (0.0)	
IA2	441 (52.4)	441 (64.2)	0 (0.0)	
IA3	3 (0.4)	3 (0.4)	0 (0.0)	
IB	98 (11.6)	0 (0.0)	98 (63.2)	
IIA	45 (5.3)	45 (6.6)	0 (0.0)	
IIB	64 (7.6)	29 (4.2)	35 (22.6)	
IIIA	20 (2.4)	4 (0.6)	16 (10.3)	
IIIB	5 (0.6)	0 (0.0)	5 (3.2)	
Unknown	3 (0.4)	2 (0.3)	1 (0.6)	

^1^ Wilcoxon rank sum test; Pearson’s Chi-squared test; Fisher’s exact test. VPI, Visceral Pleural Invasion.

**Table 3 cancers-17-03382-t003:** Univariate and multivariate analyses of cases associated with lymph node metastasis. Excluded cases of Adenocarcinoma subtypes of in situ, minimally, lepidic.

Variable	Univariable			Multivariable		
	OR ^1^	95% CI ^1^	*p*-Value	OR ^1^	95% CI ^1^	*p*-Value
**Age: (>70 vs. ≤70)**	0.83	0.64–1.08	0.20	0.85	0.65–1.12	0.26
**Sex: (women vs. men)**	0.67	0.51–0.87	0.003	0.58	0.41–0.82	0.002
**Smoking habit: (smoker vs. non-smoker)**	1.13	0.86–1.48	0.40	0.78	0.54–1.12	0.17
**Tumor location: (Left vs. Right)**	1.24	0.96–1.60	0.10	1.48	1.13–1.94	0.005
**Pathological tumor size: (>2 cm vs. ≤2 cm)**	2.06	1.59–2.68	<0.001	1.66	1.27–2.19	<0.001
**Histological type: (Ad vs. non-Ad)**	1.08	0.80–1.48	0.60	1.15	0.82–1.62	0.42
**Visceral pleural invasion: (present vs. absent)**	2.63	1.99–3.46	<0.001	2.24	1.68–2.98	<0.001

^1^ OR = Odds Ratio, Cl = Confidence Interval. Ad, Adenocarcinoma.

**Table 4 cancers-17-03382-t004:** Relationship of VPI and area of lymph node metastasis. Excluded cases of Adenocarcinoma subtypes of in situ, minimally, and lepidic ^1^.

Characteristic	Overall, *n* = 1562	Non-VPI, *n* = 1212	VPI, *n* = 350	*p*-Value ^2^ (Non-VPI vs. VPI)
**Surgical procedure, *n* (%)**				<0.001 ^2^
Segmentectomy	320 (20.5)	277 (22.9)	43 (12.3)	
Lobectomy	1242 (79.5)	935 (77.1)	307 (87.7)	
**Lymph node dissection, *n* (%)**				
Not conducted	35 (2.2)	31 (2.6)	4 (1.1)	
ND1	240 (15.4)	196 (16.2)	44 (12.6)	
ND2a-1	1069 (68.4)	839 (69.2)	230 (65.7)	
ND2a-2	218 (14.0)	146 (12.0)	72 (20.6)	
**Pathological N factor, 9th edition, *n* (%)**				<0.001 ^2^
N1	149 (9.5)	97 (8.0)	52 (14.9)	
N2a	102 (6.5)	64 (5.3)	38 (10.9)	<0.001 ^2^
N2b	32 (2.0)	14 (1.2)	18 (5.1)	<0.001 ^2^
**Lymph node metastasis to N2 area, *n* (%)**	134 (8.6)	78 (6.4)	56 (16.0)	
**Skip N2 metastasis**	38 (2.4)	24 (2.0)	14 (4.0)	0.0031 ^2^
**N2 lymph node metastasis area, *n* (%)**				<0.001 ^2^
Superior mediastinal nodes	79 (5.1)	48 (4.0)	31 (8.9)	
Inferior mediastinal nodes	38 (2.4)	24 (2.0)	14 (4.0)	
Both	16 (1.0)	6 (0.5)	10 (2.9)	
**Any N1 lymph node metastasis, *n* (%)**	245 (15.7)	151 (12.5)	94 (26.9)	<0.001 ^2^
**Single N1 station metastasis, *n* (%)**	153 (9.8)	97 (8.0)	56 (16.0)	<0.001 ^2^
**Multiple N1 station** **metastasis, *n* (%)**	92 (5.9)	54 (4.5)	38 (10.9)	<0.001 ^2^
**N1 lymph node metastasis area, *n* (%)**				
Hilar zone (#10)	44 (2.8)	21 (1.7)	23 (6.6)	<0.001 ^2^
Interlobar zone (#11)	68 (4.4)	39 (3.2)	29 (8.3)	<0.001 ^2^
Lobar zone (#12u, #12m, #12l)	157 (10.1)	98 (8.1)	59 (16.9)	<0.001 ^2^
Segmental subsegmental zone (#13, 14)	87 (5.6)	57 (4.7)	30 (8.6)	0.005 ^2^

^1^ Nine cases with missing data on lymph node metastasis were excluded. ^2^ Wilcoxon rank sum test; Pearson’s Chi-squared test; Fisher’s exact test. VPI, Visceral Pleural Invasion.

## Data Availability

The datasets generated and analyzed during the current study are not publicly available due to ethical restrictions but are available from the corresponding author on reasonable request, provided that an appropriate justification is given.

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
