# Peer review of "Visceral Pleural Invasion as a Determinant of Surgical Strategy in Non–Small Cell Lung Cancer: A Multicenter Study"

_cancers, 2025, doi:10.3390/cancers17203382_

Round 1

Reviewer 1 Report

Comments and Suggestions for Authors

This is an important and well-presented study. The large cohort and exclusion of lepidic-spectrum adenocarcinomas strengthen its validity. I recommend minor revision with the following points:

  1. Clarify in the abstract/methods that this is a retrospective study; specify mediastinal lymph node dissection details in segmentectomy cases.
  2. Provide information on Cox model assumptions and handling of missing data.
  3. Improve figure legends (include patient numbers) and standardize p-value formatting.
  4. If available, add a breakdown of local vs. distant recurrence.
  5. Moderate strong statements (e.g., “VPI appears to be” rather than “VPI is”).
  6. Minor editorial adjustments for consistency in references and style.

These refinements will further improve clarity and impact.

Author Response

For research article

Response to Reviewer 1 Comments

1. Summary

Thank you very much for taking the time to review this manuscript. Please find the detailed responses below and in track changes in the re-submitted files.

2. Questions for General Evaluation

Reviewer’s Evaluation

Response and Revisions

Does the introduction provide sufficient background and include all relevant references?

Can be improved

[Please give your response if necessary. Or you can also give your corresponding response in the point-by-point response letter. The same as below]

Are all the cited references relevant to the research?

Can be improved

Is the research design appropriate?

Yes

Are the methods adequately described?

Yes

Are the results clearly presented?

Yes

Are the conclusions supported by the results?

Yes

3. Point-by-point response to Comments and Suggestions for Authors

Comments 1:

Clarify in the abstract/methods that this is a retrospective study; specify mediastinal lymph node dissection details in segmentectomy cases.

Response 1:

We appreciate the reviewer’s suggestion. We have clarified that this is a retrospective study in the Abstract as shown below. We also specified that mediastinal lymph node dissection (ND2a) was the standard procedure in segmentectomy. Intraoperative frozen-section examination of suspicious hilar or mediastinal nodes was routinely performed, and when negative, mediastinal dissection could be curtailed at the surgeon’s discretion according to institutional protocols.

(page 1, line 31)

Methods: We conducted a retrospective comprehensive multicenter study involving 2,464 patients with pathologically confirmed NSCLC ≤3 cm who underwent complete resection at three Japanese institutions.

(page3, Paragraph 1, line 137)

Mediastinal lymph node dissection (ND2a) was the standard procedure in both lobectomy and segmentectomy. Intraoperative frozen-section examination of sus-picious hilar or mediastinal nodes was routinely performed, and when negative, me-diastinal dissection could be curtailed at the surgeon’s discretion according to the surgical strategy of each institution.

Comments 2:

Provide information on Cox model assumptions and handling of missing data.

Response 2:

Thank you for raising this important methodological issue. We apologize for the earlier ambiguity in our description. We did not use a Cox proportional hazards model for the analyses of lymph node metastasis. Time-to-event outcomes were evaluated using Kaplan–Meier curves and log-rank tests, while multivariable analysis of factors associated with lymph node metastasis was performed using logistic regression (not Cox regression). Thus, Cox proportional hazards assumptions are not applicable to our multivariable analyses. We have revised and clarified the Statistics subsection of the Methods accordingly.

Missing data were minimal (<5%) and were handled using complete-case analysis.

(page4, Paragraph 1, line187)

Survival differences between patient groups categorized by pathological lymph node metastasis and VPI status were compared using the log-rank (Mantel–Cox) test. Multivariable analyses of factors associated with lymph node metastasis were performed using logistic regression. Covariates included in the multivariable analysis were age, sex, smoking history, tumor location, pathological tumor size, histologic type, and VPI status.

Comments 3:

Improve figure legends (include patient numbers) and standardize p-value formatting.

Response 3:

Thank you for pointing this out. Figure legends of figure2 and 3 were revised to include the number of patients analyzed with standardize p-value formatting.

(page 9, line 279)

Figure 2. Overall survival curves based on VPI and lymph node metastasis. Overall survival curve excluding patients with histological subtypes: adenocarcinoma in situ (AIS), minimally in-vasive adenocarcinoma (MIA), or lepidic adenocarcinoma (Lep). In node-negative patients (n = 1278), the 5-year OS was significantly higher in those without VPI (n = 1036, 90.8%) compared to those with VPI (n = 242, 82.3%) (log-rank, p < 0.001). Among node-positive patients (n = 292), there was no significant difference in OS between VPI groups: 181 cases (70.8%) without VPI vs. 111 cases (68.0%) with VPI (log-rank p = 0.759).

(page 10, line 289)

Figure 3. Recurrence-free survival curves based on VPI and lymph node metastasis. Recur-rence-free survival curve excluding patients with histological subtypes: adenocarcinoma in situ (AIS), minimally invasive adenocarcinoma (MIA), or lepidic adenocarcinoma (Lep). Among node-negative patients (n = 1278), the 5-year RFS was 94.1% in those without VPI (n = 1036) and 72.5% in those with VPI (n = 242). In node-positive patients (n = 292), the 5-year RFS decreased to 50.6% without VPI (n = 181) and 33.7% with VPI (n = 111).

Comments 4:

If available, add a breakdown of local vs. distant recurrence.

Response 4:

We appreciate this suggestion. We do have data on recurrence sites in general and have intended to add this information. However, while recurrence site information was available, detailed anatomical data were not consistently recorded. Therefore, it was difficult to accurately distinguish between local and distant recurrences. We sincerely thank the reviewer for this valuable comment and appreciate your understanding.

Comments 5:

Moderate strong statements (e.g., “VPI appears to be” rather than “VPI is”).

Response 5:

Thank you for constructive comment. We have modified the sentences as you recommended.

(page 2, line 74)

With the increasing adoption of sublobar resection for small-sized NSCLC, recognizing that VPI appears to be associated with predominant hilar involvement and an elevated risk of skip N2 metastasis may help refine decisions on the extent of lung and lymph node resection.

Comments 6:

Minor editorial adjustments for consistency in references and style.

Response 6:

We revised the reference style throughout the manuscript and ensured consistency according to journal requirements.

4. Response to Comments on the Quality of English Language

none

5. Additional clarifications

none

Reviewer 2 Report

Comments and Suggestions for Authors

Dear Editor and Authors,

It was really a pleasure to review this research article by Dr. Nagase and his multidisciplinary colleagues from Tokyo, Japan titled "Pathological Visceral Pleural Invasion and Lymph Node Metastasis in Early-Stage Non–Small Cell Lung Cancer: A Multicenter Study"

This work deals with a subject that is of quite interest for the thoracic surgical community as it deals with an issue which we surgeons often face but is still poorly understood. Particularly as it pertains to to long term outcomes and survival! Unfortunately, viceral pleural invasion (VPI) is often a histopathological finding and not easily identified intraoperatively unless there is obvious breaking of the visceral pleura's continuity by tumour! Even then the management of such a find is unclear, if there is really something that can be done from a surgical prespective appart from continouing with the resection!!

Therefore, in this retrospective, multi-institutional study the authors draw from a pool of over 2000 patients (2,464) and identified 370 in which viceral pleural invastion was identified. They compared outcomes and survival and found that these patients had double the rate of hilar lymph node spread, a distinct pattern of lymph node metastasis with more hilar and skip metastases, reduced five year survival and generally poorer outcomes. This is not unusual, if one considers that there is potential of cancer cell shedding with VPI but this would mostly involve the pleural cavity with secondary pleural metastasis and pleural effusion development! The involvement of lymph nodes and the distinct way it behaves makes one believe that VPI is one of the charachteristic associated with aggressive tumours!

The work is actually quite well conducted! It is retrospective in nature but this is understandable in such type of analysis. The methodology followed is robust albeit I do have some questions regarding the statistics (please see comments for authors). There are clear inclusion and exclusion criteria which are the expected used in survival studies, and an adequate sample size (again I have some comments for this).

The manuscript is well written and structured and it is easily understood by the reader. It addresses and presents all sections well and presents a comprehensive study! The tables and graphs are illustrative and supportive of the work.

I do have some comments I would like addressed by the authors:

  1. Why wasn't a power analysis - sample size calculation performed to demonstrate the adequate number of cases analyzed? This should have been done pre data analysis and mining!
  2. Why wasn't propensity score matching performed to evaluate outomes more accurately and reduce bias?
  3. In terms of segmentectomy do the authors include wedge resections or only anatomical segmentectomies (segmental veins/artery/bronchus resection!!)?
  4. Why was lymph node clearance/sampling techniques differ between the two groups? Was this discrepancy which is quite significant for survival not evaluated seperately/in a multivariable model??
  5. What is the duration of recruitment for the database (period)?
  6. Where cases performed via VATS, RATS or open thoracotomy? This information seems not presented!
  7. I suggest a survival curve/analysis is performed according to tumour size as well (<2cm vs 3cm)! This was done for lymph node metastasis!

In conclusion, I enjoyed reading this study and I feel it has something (maybe significant to a degree) to offer to the clinical/surgical community. I do have some comments which I feel would improve the work!

Author Response

For research article

Response to Reviewer 2 Comments

1. Summary

Thank you very much for taking the time to review this manuscript. Please find the detailed responses below and in track changes in the re-submitted files.

2. Questions for General Evaluation

Reviewer’s Evaluation

Response and Revisions

Does the introduction provide sufficient background and include all relevant references?

Yes

[Please give your response if necessary. Or you can also give your corresponding response in the point-by-point response letter. The same as below]

Are all the cited references relevant to the research?

Yes

Is the research design appropriate?

Can be improved

Are the methods adequately described?

Yes

Are the results clearly presented?

Yes

Are the conclusions supported by the results?

Yes

3. Point-by-point response to Comments and Suggestions for Authors

Comments 1:

Why wasn't a power analysis - sample size calculation performed to demonstrate the adequate number of cases analyzed? This should have been done pre data analysis and mining!

Response 1:

We appreciate this important comment. Because this was a retrospective study using a fixed cohort, no a priori sample size calculation was performed. Instead, the adequacy of the study size was evaluated by the precision of effect estimates, expressed as the width of 95% confidence intervals for key associations.

(page 7, line 261)

Multivariate analysis identified VPI as an independent prognostic factor, along with sex, tumor location, and pathological tumor size (adjusted odds ratio = 2.24, 95% CI: 1.68–2.98, p < 0.001). The narrow confidence interval indicates sufficient precision of estimation, which was considered an alternative to a priori sample size calculation in this retrospective study.

Comments 2:

Why wasn't propensity score matching performed to evaluate outomes more accurately and reduce bias?

Response 2:

We thank the reviewer for this valuable suggestion. However, propensity score matching is not applicable in the present study. Propensity methods are most suitable when comparing treatment strategies (e.g., lobectomy vs. segmentectomy) based on preoperative covariates under the framework of emulating a prospective trial. In contrast, visceral pleural invasion (VPI) is a postoperative pathological finding and therefore unsuitable as a matching variable for surgical allocation.

To address confounding in survival analysis, we instead performed subgroup analyses restricted to node-negative patients, since nodal status is a well-established prognostic determinant. Within this homogeneous cohort, the adverse prognostic impact of VPI remained evident, supporting the robustness of our results. In addition, multivariable Cox regression models included other key covariates to further account for potential imbalances.

We sincerely appreciate the reviewer’s understanding of this point and thank you for the constructive feedback. 

Comments 3:

In terms of segmentectomy do the authors include wedge resections or only anatomical segmentectomies (segmental veins/artery/bronchus resection!!)?

Response 3:

Thank you for the remark. Segmentectomy only included anatomical segmentectomies.

(page 3, line 134)

Patients were excluded if they had tumors larger than 3 cm, a pathological diagnosis of small cell lung cancer, or had undergone procedures other than lobectomy or segmentectomy. Wedge resections were not included in this analysis.

Comments 4:

Why was lymph node clearance/sampling techniques differ between the two groups? Was this discrepancy which is quite significant for survival not evaluated seperately/in a multivariable model??

Response 4:

We agree that the extent of lymph node dissection may influence the detection of nodal metastasis. In the main analysis, we reported raw associations, as lymph node dissection is also determined intraoperatively and can be considered part of the causal pathway. Nonetheless, we performed an additional multivariable analysis including the extent of lymph node dissection (ND1, ND2a-1, ND2a-2) as a covariate. VPI remained significantly associated with nodal metastasis, indicating that the findings were not explained solely by differences in surgical technique.

(page 7, line 266)

To further validate the findings, we additionally performed multivariate analysis including the extent of lymph node dissection, and confirmed that VPI remained an independent prognostic factor; the results were consistent (data not shown).

Comments 5:

What is the duration of recruitment for the database (period)?

Response 5: Thank you for the comment. Recruitment covered cases from January 2010 to December 2019. We have revised the manuscript as follows.

(page 3, line 131)

A total of 2,464 patients with surgically resected NSCLC tumors measuring 3 cm or less in pathological whole tumor size were included between January 2010 and December 2019. Data were collected from three institutions, Hiroshima University, Kanagawa Cancer Center, and Tokyo Medical University.

Comments 6:

Where cases performed via VATS, RATS or open thoracotomy? This information seems not presented.

Response 6:

We appreciate the reviewer’s interest in the surgical approach. Unfortunately, the database used for this analysis did not include detailed information regarding the surgical method (VATS, RATS, or thoracotomy). Therefore, we are unable to provide this information. We sincerely thank the reviewer for this valuable comment and appreciate your understanding.

Comments 7:

I suggest a survival curve/analysis is performed according to tumour size as well (<2cm vs 3cm). This was done for lymph node metastasis.

Response 7:

Thank you so much for your important comment. We have added the survival curves stratified by tumor size and VPI, together with corresponding descriptions in the Results.

(page 7, line 250)

In patients with tumors <2 cm, there was no significant difference in overall survival between those with and without VPI (5-year OS: 85.1% vs. 89.1%, log-rank, p = 0.128). In contrast, among patients with tumors 2 to 3 cm, survival was significantly worse in those with VPI compared to those without VPI (5-year OS: 73.1% vs. 86.8%, log-rank, p < 0.001)(see Supplementary Figure S3). Regarding recurrence-free survival, patients with tumors <2 cm with VPI had significantly worse outcomes compared to those without VPI (5-year RFS: 75.0% vs. 90.5%, log-rank, p < 0.001). Similarly, in tumors 2 to 3 cm, recurrence-free survival was significantly poorer in patients with VPI than in those without VPI (5-year RFS: 50.7% vs. 84.7%, log-rank, p < 0.001)(see Supplementary Figure S4).

5. Additional clarifications

none

Reviewer 3 Report

Comments and Suggestions for Authors

The clinical implications of visceral pleural invasion (VPI) in non-small cell lung cancer (NSCLC), particularly in small NSCLC, were examined. A multicenter analysis of data from 2,464 patients indicated that VPI correlated with an elevated lymph node metastasis rate and distinct metastatic patterns, including increased risks of hilar spread and skipped N2 metastasis. Furthermore, VPI significantly decreased the five-year recurrence-free survival rate. The study observed that adenocarcinomas with lepidic features demonstrated lower risks of VPI and lymph node metastasis, highlighting the biological heterogeneity of small NSCLCs. Consequently, VPI serves not only as a pathologic descriptor but also as a clinical indicator for assessing aggressive metastatic behavior, potentially aiding in the refinement of surgical and staging strategies. It is recommended that the author revise the manuscript in accordance with the following comments.

  1. It is recommended that the author allocate additional space in the introduction to provide comprehensive background information on non-small cell lung cancer.
  2. The following references are highly relevant to the author's topic and are recommended for citation.

[1] Wu YL, Lu S, Zhou Q, Zhang L, Cheng Y, Wang J, et al. Expert consensus on treatment for stage III non-small cell lung cancer. Med Adv. 2023; 1(1): 3–13. https://doi.org/10.1002/med4.7

[2] A. Gu, J. Li, M.-Y. Li, Y. Liu, Patient-derived xenograft model in cancer: establishment and applications. MedComm, 2025, 6, e70059. DOI: 10.1002/mco2.70059

[3] Dai Y, Tian X, Ye X, Gong Y, Xu L, Jiao L. Role of the TME in immune checkpoint blockade resistance of non-small cell lung cancer. Cancer Drug Resist. 2024;7:52. http://dx.doi.org/10.20517/cdr.2024.166

  1. In order to mitigate the constraints imposed by the retrospective character of this study, we propose a prospective study as a means to confirm the correlation between VPI, LN metastasis, and Skip N2 metastasis.
  2. It is strongly suggested that a standardized definition and metric for VPI be adopted to facilitate meaningful comparisons of data across various centres.
  3. Through the utilization of imaging techniques and artificial intelligence tools in actual surgical procedures, there is potential evidence to support the preoperative prediction of VPI. This can be further substantiated by assessing their impacts and effectiveness.
  4. It is advised that analyses be conducted separately based on the specific tumor location, such as the lung lobe or lung segment. Additionally, comprehensive descriptions of lymph node involvement should be incorporated to fully comprehend the implications of VPI.
  5. It is suggested that in addition to histological subtypes, other factors influencing the incidence of VPI should be examined. These may include smoking history, age, and gender.

Author Response

For research article

Response to Reviewer 3 Comments

1. Summary

Thank you very much for taking the time to review this manuscript. Please find the detailed responses below and in track changes in the re-submitted files.

2. Questions for General Evaluation

Reviewer’s Evaluation

Response and Revisions

Does the introduction provide sufficient background and include all relevant references?

Must be improved

[Please give your response if necessary. Or you can also give your corresponding response in the point-by-point response letter. The same as below]

Are all the cited references relevant to the research?

Must be improved

Is the research design appropriate?

Yes

Are the methods adequately described?

Yes

Are the results clearly presented?

Yes

Are the conclusions supported by the results?

Yes

3. Point-by-point response to Comments and Suggestions for Authors

Comments 1:

It is recommended that the author allocate additional space in the introduction to provide comprehensive background information on non-small cell lung cancer.

Response 1:

We appreciate the reviewer’s interest. The following sentences have been added in the introduction.

(page 2, line 83)

Lung cancer remains one of the leading causes of cancer-related mortality worldwide, with prognosis remaining generally unfavorable(引用:https://doi.org/10.3322/caac.21871). Despite recent advances in perioperative systemic therapies that have raised expectations for improved outcomes, even small, early-stage non–small cell lung cancer (NSCLC) can harbor occult lymph node metastasis [2,3].

Comments 2:

The following references are highly relevant to the author's topic and are recommended for citation.

[1] Wu YL, Lu S, Zhou Q, Zhang L, Cheng Y, Wang J, et al. Expert consensus on treatment for stage III non-small cell lung cancer. Med Adv. 2023; 1(1): 3–13. https://doi.org/10.1002/med4.7

[2] A. Gu, J. Li, M.-Y. Li, Y. Liu, Patient-derived xenograft model in cancer: establishment and applications. MedComm, 2025, 6, e70059. DOI: 10.1002/mco2.70059

[3] Dai Y, Tian X, Ye X, Gong Y, Xu L, Jiao L. Role of the TME in immune checkpoint blockade resistance of non-small cell lung cancer. Cancer Drug Resist. 2024;7:52. http://dx.doi.org/10.20517/cdr.2024.166

Response 2:

Thank you so much for the helpful comment. As the reviewer suggested, we have added the recommended article for citation as follows.

(page 2, line 84)

Despite recent advances in perioperative systemic therapies that have raised expectations for improved outcomes, even small, early-stage non–small cell lung cancer (NSCLC) can harbor occult lymph node metastasis [2,3].

Comments 3:

In order to mitigate the constraints imposed by the retrospective character of this study, we propose a prospective study as a means to confirm the correlation between VPI, LN metastasis, and Skip N2 metastasis.

Response 3:

We fully agree with the reviewer that a prospective study would be the optimal approach to validate the relationship between VPI, lymph node metastasis, and skip N2 metastasis. However, given the design of the present work, we believe our large multicenter retrospective cohort provides valuable and clinically relevant insights. Therefore, while we acknowledge this important point, we would like to present the current findings as they are, with the limitations clearly noted in the Discussion.

(page 14, line 419)

Future prospective studies are warranted to confirm the associations identified between VPI, lymph node metastasis, and skip N2 metastasis, and to elucidate their clinical im-plications for surgical decision-making.

Comments 4:

It is strongly suggested that a standardized definition and metric for VPI be adopted to facilitate meaningful comparisons of data across various centres.

Response 4:

We sincerely appreciate the reviewer’s suggestion to adopt a standardized definition for VPI. In response, we have clarified in the Methods section.

(page 3, line 163)

VPI was defined as tumor invasion beyond the elastic layer of the pleura, classified into four categories: PL0 (no invasion beyond the elastic layer), PL1 (invasion beyond the elastic layer but not reaching the pleural surface), PL2 (invasion to the pleural surface), and PL3 (invasion into the parietal pleura or chest wall). This standardized classification was applied to all surgically resected tumors in this study [16].

Comments 5:

Through the utilization of imaging techniques and artificial intelligence tools in actual surgical procedures, there is potential evidence to support the preoperative prediction of VPI. This can be further substantiated by assessing their impacts and effectiveness.

Response 5: Thank you for this important suggestion. We have added references in the discussion about recent evidence on imaging and AI-based prediction of VPI.

(page 12, line 353)

Recent investigations have focused on the preoperative and intraoperative prediction of VPI using imaging modalities and AI-based approaches [24-30].

Comments 6:

It is advised that analyses be conducted separately based on the specific tumor location, such as the lung lobe or lung segment. Additionally, comprehensive descriptions of lymph node involvement should be incorporated to fully comprehend the implications of VPI.

Response 6:

We sincerely appreciate the reviewer’s valuable suggestion to conduct analyses according to tumor location and to provide detailed descriptions of lymph node involvement. In response, these analyses are presented in the Supplementary Figure for clarity and completeness.

Comments 7:

It is suggested that in addition to histological subtypes, other factors influencing the incidence of VPI should be examined. These may include smoking history, age, and gender.

Response 7:

We appreciate this suggestion. These factors were examined in our cohort. As shown in Table 1, VPI‐positive cases were more frequently male (68.8% vs. 57.6%, p < 0.001) and smokers (73.4% vs. 63.0%, p < 0.001), whereas age did not differ materially between groups (median 68 vs. 69 years, p = 0.067). We have added a clarifying sentence in the Results to make this explicit.

(page 4, line 206)

In this cohort, VPI-positive tumors occurred more often in men and smokers, while age distributions were similar between groups.

4. Response to Comments on the Quality of English Language

None

5. Additional clarifications

none

Round 2

Reviewer 2 Report

Comments and Suggestions for Authors

Dear Editor and Authors,

Thank you for asking me to re-evaluate this revised manuscript. I was happy to see that the authors have addressed all the issues raised by myself and other reviewers and have made appropriate changes/edits. Therefore,I am happy to now recommend its acceptance for publication.

Kind regards to all and congradulation to the authors for a very nice work.